Application of fenugreek in ruminant feed: implications for methane emissions and productivity

Zeng Xiangbiao 1
Chen Yiwen 1
Li Wenjuan wjli@shou.edu.cn 1
Liu Shijun liushijun@moogonutrition.com 2
1 College of Fisheries and Life Science, Shanghai Ocean University , Shanghai , China
2 Shanghai Mugao Biotechnology Co., Ltd , Shanghai , China
Kormas Konstantinos
Electronic publication date: 2024 Jan 31
Publication date: 2024
Volume: 12
Electronic Location ID: e16842
Received 2023 Oct 27; Accepted 2024 Jan 5
Copyright: ©2024 Zeng et al.
Copyright year: 2024
Copyright holder: Zeng et al.
License: This is an open access article distributed under the terms of the Creative Commons Attribution License, which permits unrestricted use, distribution, reproduction and adaptation in any medium and for any purpose provided that it is properly attributed. For attribution, the original author(s), title, publication source (PeerJ) and either DOI or URL of the article must be cited.
License URL: https://creativecommons.org/licenses/by/4.0/

Keywords: Fenugreek, Ruminant, CH4, Productivity, Rumen fermentation, Bioactive secondary metabolites

Funding: The National Key R&D Program of China 2018YFD0901406 The National Natural Science Foundation of China 31872565 This study was financially supported by the National Key R&D Program of China (2018YFD0901406), the National Natural Science Foundation of China (31872565). The funders had no role in study design, data collection and analysis, decision to publish, or preparation of the manuscript.

==============================
Background

Human demand for meat and dairy products will increase as a result of economic development and population growth, and the farming of ruminants, such as cattle and sheep, will also increase. Methane (CH4) emission from the enteric fermentation of ruminant livestock is a major source of greenhouse gas emissions and a significant contributor to global warming. Meanwhile, growth performance is often limited and animals are more vulnerable to diseases in high-density, intensive farming, greatly reducing livestock productivity, so developing ways to reduce CH4 emissions and improve ruminant productivity has become a research hotspot. Studies have reported that fenugreek (Trigonella foenum-graecum L.) as feed additives have the potential to reduce ruminant methane and improve the productivity. However, systematic reviews of such studies are lacking.

Methodology

In this review, databases of Google Scholar, Web of Science, PubMed, Scopus and Science Direct were used for the literature search. The initial keywords search was fenugreek or Trigonella foenum-graecum L. For more focused search, we added terms such as methane, rumen fermentation, growth, milk production and antioxidants. All were done for ruminants. The literature that conforms to the theme of this article is selected, summarized, and finally completed this article.

Results

By regulating the rumen microbiome (suppressing protozoans, methanogenic bacteria, and fungi), fenugreek can lower CH4 emissions according to many in vitro anaerobic fermentation experiments. Fenugreek secondary metabolites (saponins and tannins) are responsible for this impact, but it is still unclear exactly how they work. Therefore, more long-term in vivo experiments are needed to verify its efficacy. Fenugreek is also rich in alkaloids, amino acids, flavonoids, saponins and phenolic acids. These compounds have been shown to have beneficial effects on ruminant growth, lactation, and total antioxidant capacity. Therefore, fenugreek has a great opportunity to develop into a new green feed additive.

Conclusions

This review provides a summary of the effect of fenugreek and its bioactive compounds on rumen fermentation, CH4 emissions and production performance by ruminants. In addition, based on the available data, the possible biochemical pathway of fenugreek to reduce CH4 emissions in ruminants was described. Overall, the livestock feed industry has the opportunity to develop natural, environmentally-friendly feed additives based on fenugreek.

Introduction

Concentrations of methane (CH4) in the atmosphere have soared since 2007, the recent rapid rise in global CH4 concentrations is primarily biogenic, most likely due to agriculture, according to the new data (Saunois et al., 2016). Ruminants play an important role in sustainable agriculture, among which is the transformation of renewable resources into human-edible food (Ripple et al., 2014). But their production processes also serve as a source of greenhouse gases (GHG) for various supply chain activities. One of the main ways ruminants release CH4 is through enteral fermentation, which is the microbial fermentation of feed in the gastrointestinal tract (Thorpe, 2009). The main agricultural source of CH4 and the third-largest source of greenhouse gas emissions, behind energy and industry, is enteral fermentation (Frank et al., 2019). Data shows that the global livestock sector contributes roughly 14.5% of all anthropogenic GHG emissions, with ruminants accounting for close to 12% of these emissions and approximately 44% of these emissions are CH4 (Fig. 1) (Gerber et al., 2013). Ruminant GHG emissions are thought to be responsible for around US$0.679 trillion and US$13 billion in damages to the environment and human health, respectively (Goedkoop et al., 2009). Economic development and population growth will further expand human demand for meat and dairy products. By 2050, there will be a 70% increase of global food demand, which is expected to result in a 30–40% increase in agricultural emissions (Hristov et al., 2013). This runs counter to the European Council Directorate-General for Climate Action goal, which states that developed countries must reduce GHG emissions by 80–90% by 2050, compared with 1990 levels (Sanchez-Martinez, Benedetti-Fasil & Christensen, 2019). There is tremendous pressure on food production systems to meet demand while also being environmentally friendly, sustainable, and climate-smart (Hristov et al., 2013). Therefore, novel solutions and more effective CH4 mitigation strategies are required for the sustainable development of ruminant farming (Beauchemin et al., 2020; Tseten et al., 2022).

Figure 1 Global GHG emissions of livestock industry.

(A) Anthropogenic GHG emissions worldwide. (B) GHG emissions contributed by the ruminants.

A further challenge in reducing CH4 emissions from the agricultural sector is the growing demand for milk and meat (Eisler et al., 2014). Ruminant livestock and their products are an important food in human nutrition, especially in less developed countries. How to find a new method that can meet the high yield of high-quality protein, control CH4 emissions and maximize productivity (lactation, growth, immunity and other properties) under the conditions of high density and intensive culture is very vital (Tseten et al., 2022). While numerous technical alternatives are available, implementation of these interventions may be hampered by significant infrastructure and precision nutrition strategy investment costs (Tseten et al., 2022). Adding some plants that are rich in condensed tannins and saponins to feed appears to have the potential to mitigate CH4 emissions and increase productivity due to the direct effect of diet on rumen fermentation patterns (Ku-Vera et al., 2020). Farmers may decide to employ the approach of supplementing animal feed since it has several advantages, including the promotion of growth, improvement of animal health, reduction of GHG emissions of livestock, and sustainability (Du et al., 2018). The addition of supplements to feed or diets may improve nutritional absorption and retention. Since the scientific community and the animal protein community are moving away from the use of antibiotics as growth promoters due to microbial resistance, alternatives are needed (Manyi-Loh et al., 2018). Researchers are becoming more interested in using phytogenous products as feed supplements in animal production (Adegbeye et al., 2019).

Fenugreek (Trigonella foenum-graecum L.), named Trigonella for its triangular-shaped, yellowish flowers and leaves, is an annual herb of the legume family that is native to the Mediterranean region and Indian subcontinent (Syed et al., 2020). Farmed for food, spices, animal feed, and herbal medicine, fenugreek is grown extensively in Asia, Africa, and Europe (Ahmad et al., 2016). Fenugreek is also a dryland crop, the ability of fenugreek to grow in desert regions typified by high temperatures and water scarcity makes it a good choice for future phytogenous feed additives (Wijekoon et al., 2021). Past research on fenugreek has focused on its medicinal functions and nutritional value; products derived from its seeds and leaves have been used extensively in various pharmaceuticals, nutritional products, and clinical treatments (Singh et al., 2022). Fenugreek contains a variety of chemical components, such as saponins, phenolic acid derivatives, steroids, terpenes, alkaloids, flavonoids, amino acids, and fatty acids and their derivatives (Fig. 2) (Singh et al., 2022), which have recently attracted great research interest and are considered to have the potential to mitigate CH4 emissions of ruminants at therapeutically-effective dosages. Dey (2015) found in the simulated rumen fermentation experiment in vitro that incubation of 30 ml of buffered rumen fluid with 2 ml methanol extract of fenugreek leaves for 24 h significantly reduced the total gas and methane production. Kumar et al. (2016) has similar results. Meanwhile, fenugreek as a growth promotor, lactagogue, immunostimulant and antioxidative agent application in animal feed has been widely recognized. Adding 0.04% fenugreek in the diet can significantly improve the growth and plasma biochemical indexes, and enhance the immunity and antioxidant of juvenile blunt snout bream (Yu et al., 2019). This has also been demonstrated in animals such as chickens, rabbits and sheep (Abdel-Wareth et al., 2021; El-Tarabany et al., 2018; Paneru et al., 2022; Yang et al., 2022). Fenugreek has been used as a galactologue since ancient times in nations such as China, Egypt and Persia, owing to the role of trigonelline. It has also been demonstrated to increase lactation in animals such as mice, cattle, and sheep (Daddam et al., 2023; Muddathir, 2012; Sevrin et al., 2019). Therefore, it has the potential to be a new type of green product (Acharya, Thomas & Basu, 2008; Huang et al., 2022; Sahoo et al., 2020).

Figure 2 Chemical composition of fenugreek.

This review aims to provide insight into fenugreek in terms of enteric CH4 mitigation efficacy, growth-promoting and lactation-promoting functions in ruminants, possible mode of action, and safety deriving data from reported research data. It could provide clear guidelines for fenugreek application, either as a feed additive or forage supplement. We believe that this article will be of interest to professionals involved in greenhouse gas mitigation, the development of novel feed products for livestock, plant secondary metabolites, and new green antibiotic alternatives.

Survey methodology

The databases of Google Scholar, Web of Science, PubMed, Scopus and Science Direct were used for the literature search. Articles with unavailable abstracts and nonpeer-reviewed were excluded. The major literature bibliography was also used to search for other connected publications. The initial keywords search was fenugreek or Trigonella foenum-graecum L. For more focused search, we added terms such as methane, rumen fermentation, growth, milk production, antioxidants, all for the sake of the ruminant. The articles that came up in the first search were then carefully reviewed by examining the abstract and scanning the entire text. The literature that conforms to the theme of this article is selected, summarized, and finally completed this article.

Fenugreek as a forage crop

Fenugreek is a bloat-free legume that serves as a medicine or functional food, as well as a fodder crop for livestock and is predominantly cultivated in Southern Europe, Asia and North Africa (Acharya, Thomas & Basu, 2008). Fenugreek has great potential as a substitute forage crop, based on seed yield, nutritional value and yearly output. Alfalfa (Medicago sativa L.) is one of the many herbaceous legumes that are currently used as high-nutrient fodder crops. The nutritional content of fenugreek for fodder is higher, or equal to that of alfalfa, at all growth stages (Islam et al., 2017). When alfalfa and fenugreek were compared as fodder for cows, there was no significant difference in the digestibility of the two feeds, however, the fenugreek-fed cows had a lower food intake, but a higher milk output than the alfalfa-fed cows (Alemu & Doepel, 2011). Fenugreek can be used for short-term rotations, because it is rich in crude protein and does not cause flatulence of ruminants, so it has the potential to be used as forage for young ruminants (Nedelkov et al., 2019). Early-harvested whole fenugreek had higher crude protein and fiber concentrations than first-flowering alfalfa, whereas the fiber content of mature fenugreek was lower than alfalfa and their crude protein contents were similar (Niu et al., 2021). Fenugreek is a desirable feed crop in the Canadian cattle sector, because it contains compounds (e.g., diosgenin) that support animal growth and it appears that cows prefer the taste of fenugreek (Harper et al., 2016).

With yields of up to 1,300 kg/hectare, fenugreek seed has the potential to be used in animal feed, however yields vary greatly depending on the variety (Basu et al., 2009). Unfortunately, genetic diversity amongst fenugreek genotypes has rarely been estimated. Fenugreek fodder yield is strongly correlated with climate, being higher in semi-arid areas and lower with frequent rainfall; fenugreek production in southern Alberta during the wet season only produces an average yield of 5.8 tons of dry matter per hectare, which may limit its viability (Mir et al., 1997). Fenugreek is a legume that fixes atmospheric nitrogen 283 kg N ha−1 year−1, reducing the requirement for nitrogen fertilizer for succeeding crops and boosting the growth of subsequent crops in the rotation (Zandi et al., 2015). Fenugreek is a dryland crop, which requires very little water, therefore growing it can reduce irrigation expenses, water consumption, eutrophication of surface water sources and depletion of groundwater sources (Solorio-Sánchez et al., 2014). Fenugreek can also grow adequately in slightly alkaline, or marginally saline soils, however, its ability to grow on saline soils has only been partially assessed (Ahmad et al., 2016).

Overall, fenugreek has great potential as a fodder crop, but its properties also restrict its growth and use, so more research is needed to enhance its crop yield, stability, and performance.

Fenugreek and ruminant CH4 emissions

An introduction to rumen CH4 synthesis

The stomach of ruminants is composed of four interconnected compartments, called rumen, reticulum, omasum and abomasums from front to back, of which the most important and most functional is the rumen (about 80% of the total volume) (Morgavi et al., 2013). The rumen is an anaerobic, methanogenic environment with a diverse and dynamic microbial community. It consists of bacteria (1010 to 1011 organisms/ml), archaea (108 to 109 organisms/ml), protozoa (105 to 106 organisms/ml), fungi (103 to 104 organisms/ml) and an as yet largely uncharacterized virome (Newbold & Ramos-Morales, 2020). They actively participate in the degradation of feed elements that are lignocellulosic (Lourenco, Ramos-Morales & Wallace, 2010). After ruminant ingestion, the carbohydrates are degraded into monomers by rumen microorganisms, which are then fermented into volatile fatty acids (VFAs). The VFAs contain acetate, butyrate and propionate, along with methanol, methylamine and minor VFAs of valerate, iso-butyrate. These VFAs are either digested in the rumen wall to produce CH4 or are absorbed along the ruminal epithelium and used as the ruminant’s energy source (Morgavi et al., 2012; Newbold & Ramos-Morales, 2020).

Methanogens transform the primary fermentation products into CH4 and generate energy. CH4 is mainly produced in ruminants in three ways: (1) Through the action of microbial enzymes and coenzymes, CO2 is reduced with H2 to produce CH4; this is the main pathway of rumen CH4 production. In this method, 4 moles of H2 are used to produce 1 mole of CH4. The chemical reaction for synthesizing CH4 is CO2 + 4H2 → CH4 + 2H2O; (2) using VFAs, such as formic, acetic, propionic and butyric acid, as substrates to synthesize CH4. Formate is a common substitute for H2 for many hydrogenotrophic rumen methanogenic bacteria, accounting for up to 18% of the total CH4 production in the rumen (Hungate et al., 1970); (3) using methyl compounds, such as methanol and ethanol, as substrates to synthesize CH4 (DiMarco, Bobik & Wolfe, 1990; Newbold & Ramos-Morales, 2020).

Secondary metabolites of fenugreek that reduce CH4 emissions and the potential effect on biochemical pathways

Secondary metabolites of fenugreek.

Secondary metabolites are highly bioactive and have been widely used to improve rumen fermentation and also have the potential to influence the growth rate of rumen microbial populations, thereby reducing CH4 emissions from the ruminant gut (Ku-Vera et al., 2020). The potential of tannin (Aboagye & Beauchemin, 2019; Min et al., 2020), saponin (Goel, Makkar & Becker, 2008a; Patra & Saxena, 2009), essential oil (Benchaar et al., 2008) and flavonoid compounds (Olagaray & Bradford, 2019) to mitigate CH4 in the intestinal tract of ruminants has all been evaluated. Secondary metabolites are quite rich in fenugreek, especially in its leaves and seeds, including steroidal saponins, flavonoids, alkaloids, amino acids, phenolic acids and tannins (Table 1) (Omezzine et al., 2017). Figure 3 shows the structural diversity of the compounds isolated from fenugreek. Saponins are high molecular weight glycosides that usually occur as glycosides or polycyclic triterpenes of steroids. Saponins are structurally characterized by either a hydrophobic triterpenoid ring or a steroidal skeleton (Venkata et al., 2017). In the main components of fenugreek seeds, the content of steroidal saponins is about 6.5%. At present, dozens of saponins have been isolated from fenugreek, of which the diosgenin element dominates, which is believed to be the precursor of many sex hormones (Srinivasan, 2006). Flavonoids are polyphenols with the C6-C3-C6 skeleton, which are mainly distributed in seeds, stems and leaves, and more than 10 kinds of flavonoids have been isolated and extracted from fenugreek (Singh et al., 2022). The majority of flavonoids in fenugreek are present as complex glycosides, conjugating with carbohydrates via O- and C-glycosidic bonds, according to the phytochemical examination of the plant (Venkata et al., 2017). It displays anti-inflammatory, antioxidant, and antibacterial properties and interferes with different bactericidal agents, and has the potential to improve animal welfare (Cushnie & Lamb, 2011). Tannins are a class of water-soluble polyphenolic polymers, which have an important role in controlling parasites and regulating microorganisms (Naumann et al., 2017). Numerous significant phenolic compounds with antioxidant, anticancer, and antibacterial effects, including p-coumaric acid, caffeic acid, and chlorogenic acid, have been extracted from fenugreek (Ahmad et al., 2016).

Table 1 Biologically active constituents of fenugreek and their pharmacological efficacy.

Active substances	Efficacy	References	
Alkaloids	Hypoglycemic, antioxidant, neuroprotective, anti-migraine, antibacterial, antitumor	Zhou, Chan & Zhou (2012)	
Amino Acids	Insulin stimulating activity, increasing insulin sensitivity	Ahmad et al. (2020)	
Saponins	Anti-bacterial, anti-diabetes, heart protection, anti-cancer	Spandan et al., (2017)	
Flavonoids	Antioxidant, anti-bacterial, hypoglycemic	Wang et al. (2019)	
Phenolic acids and derivatives	Antioxidant, anti-cancer, anti-bacterial	Yoshikawa et al. (1997), Grusak (2003)	
Others	Regulatory functions	Hasanzadeh et al. (2010)	

Figure 3 Structures of fenugreek phytochemicals.

(A) Alkaloids. (B) Saponins. (C) Flavonoids. (D) Phenolic acids and derivatives.

Effect of fenugreek on rumen microbiota.

Rumen microorganisms include ciliate protozoa, anaerobic fungi, anaerobic bacteria and archaea, which all contribute to rumen fermentation; 90% of CH4 emitted from ruminants is produced by rumen microorganisms (Tapio et al., 2017). Because of the symbiotic interaction between archaea and protozoa as well as the presence of active colonies of methanogenic archaea on their exterior surfaces, protozoa are indirectly engaged in the generation of CH4 (Morgavi et al., 2013; Newbold & Ramos-Morales, 2020). Starch, cellulose, hemicellulose, pectin, and soluble sugars are used by protozoa to create metabolic H2 and VFA, which are needed by archaea to create CH4 (Newbold & Ramos-Morales, 2020). Saponin-containing plants can inhibit CH4 emissions by reducing the protozoal population and changing rumen fermentation conditions (Morgavi, Jouany & Martin, 2008). For example, in an in vitro rumen simulation, adding 11.54 mg fenugreek saponins to a buffer medium containing rumen microorganisms (40 ml), it was found that the antigen-parasite activity was the highest at this time, reaching 39%. Protozoa concentration decreased from 19.5 to 11.9 (×104) per ml and the number of fungi and methanogens also declined (Goel, Makkar & Becker, 2008a). Adding 48, 54, 60, or 66 mg fenugreek plant material to 200 mg vetch-oat hay, as the basal feed in an in vitro rumen simulation, reduced protozoan numbers by 45, 35, 72 and 67%, respectively (Arhabbr, Ablabr & Zitouni, 2014). This was mainly attributed to the strong anti-protozoal activity of saponins. The lipid membranes of bacteria can form compounds with saponins, increasing their permeability and creating an imbalance that triggers microbial breakage (Kumar et al., 2016; Goel & Makkar, 2012; Ibidhi & Salem, 2022). Protozoa and methanogens exhibit endo- or exo-symbiosis in the rumen, so CH4 production by the combination of protozoa and methanogens depends on the extent of symbiosis between methanogens and protozoa, as well as the methanogenesis rate of each methanogenic strain (Machmuller, Soliva & Kreuzer, 2003). Suppressing protozoa reduces their H2 production; methanogenic bacteria need H2 to reduce CO2, so this reduces CH4 production (Tapio et al., 2017). However, there are other reports that although saponins suppress the growth of protozoa and methanogens, they have no effects on CH4 production. This could be due to the slow binding between protozoa and methanogens, or the increased metabolism of methanogens after the addition of saponins, resulting in increased CH4 production (Patra, Kamra & Agarwal, 2006). The effect of saponins on protozoans may be temporary because bacteria can degrade saponins into sapogenins, which do not affect protozoans. Fenugreek saponins also inhibit fungal populations, which, like protozoa, interact closely with methanogens and are thought to promote CH4 production (Goel, Makkar & Becker, 2008a). Li (2015) reported that by adding 50 mg fenugreek extract to 500 mg fermentation substrate, the number of fiber-degrading bacteria Butyrivibrio fibrisolvens and Fibrobacter succinogenes are significantly reduced, while methanogenic bacteria and protozoa are not significantly different from the control group. Butyrivibrio fibrisolvens and Fibrobacter succinogenes can not only degrade cellulose, but also perform biohydrogenation, which can reduce unsaturated fatty acids in the rumen to conjugated linoleic acid. CH4 production may be influenced by the fatty acid biohydrogenation pathway, suggesting that this may be an alternative pathway for CH4 reduction in fenugreek (Hassan et al., 2020).

Fenugreek is a superior forage plant that significantly reduces CH4 emissions in simulated in vitro rumen fermentations, which may be related to the abundant tannin in fenugreek (Niu et al., 2021). Naseri, Hozhabri & Kafilzadeh (2013) reported that the tannin content of fenugreek seed was 3.8 g/kg, more than twice that of asparagus root (1.5 g/kg), and much higher than that of alfalfa hay (0.4 g/kg). In a study involving 17 plants containing polyphenols, a negative correlation was observed between total phenol, total tannic acid, or tannin activity, and CH4 production, which indicates the potential CH4-reducing ability of fenugreek (Jayanegara et al., 2009). Tannins can inhibit the interactions between methanogens and protozoans, by binding to adhesion proteins, or part of the cell wall, decreasing interspecies H2 transfer and inhibiting the growth of methanogenic bacteria, thereby reducing the numbers of methanogens and protozoa, directly or indirectly (Doepel et al., 2012). Tannins can also inhibit the production of CH4 by reducing the activity of fiber-degrading bacteria; the H2 provided by fiber degradation is the substrate for the conversion of pyruvate to acetate in the CH4 biosynthesis pathway (Tavendale et al., 2005). Min et al. (2005) showed that concentrated tannin reduced the number of fiber-degrading bacteria in the rumen, thereby reducing the microbial digestibility of dietary fiber polysaccharides and CH4 production. However, the mechanism of action by which fenugreek tannins inhibit CH4 production is not clear. It may influence microbial fermentation through antibacterial activity against specific rumen microbial strains, or selectively regulate specific rumen bacteria and fungi.

Fenugreek flavonoids are effective antibacterial agents, inhibiting gram-positive bacteria by interfering with the formation of cell walls, nucleic acids, or membranes (Olagaray & Bradford, 2019). Oskoueian, Abdullah & Oskoueian (2013) found that quercetin can reduce the proliferation of ciliate protozoa and methanogens in vitro. In addition, plant essential oils generally have antibacterial activity against a variety of rumen microorganisms, can effectively reduce the number of methanogenic bacteria in rumen fluid, and are most effective against gram-positive bacteria (Szumacher-Strabel & Cieslak, 2010). However, there has been no report to date on the effect of fenugreek essential oil on rumen microorganisms in vivo.

Effects of fenugreek on volatile fatty acid production.

The anaerobic environment of the rumen results in incomplete fatty acid metabolism and the final products are VFAs and CO2, so analysis of VFAs can indirectly reflect the status of rumen fermentation (Ugbogu et al., 2019). Adding 10% fenugreek to the alfalfa hay substrate increased the metabolic rate of dry matter (DM) in a rumen simulation, decreasing the VFAs concentration and total gas production (CH4, NH3, CO2). This was attributed to fenugreek contributing a higher proportion of substrate to microbial biomass production, indicating its effectiveness in influencing the supply of metabolized DM to rumen microbes (Naseri, Hozhabri & Kafilzadeh, 2013). Fenugreek- and alfalfa-based substrates did not result in differences in pH, or VFAs production, but fenugreek increased the proportion of propionic acid produced by an in vitro rumen simulation (Mir et al., 1997). Goel, Makkar & Becker (2008a) also observed a slight increase in the proportion of propionic acid in the VFAs produced at low fenugreek concentration (5.6 mg/40 ml). It’s indicated that fermentative FG reduces CH4 uptake by H2 or increases propionic acid uptake by H2 by decreasing methanogenic activity by reducing CH4 production (Mir et al., 1997; Tapio et al., 2017).

Propionic acid is the only gluconeogenic VFA that has the ability to enhance the efficiency of metabolizable energy utilization throughout the animal. In general, the ratio of acetic to propionic acid affects CH4 production, and the lower the ratio, the lower the CH4 production (Moorby & Fraser, 2021). This is because propionic acid provides an alternative pathway to metabolize H2, away from CH4 production (Ku-Vera et al., 2020). The effect of fenugreek on VFAs appears to be mediated by saponins, which can inhibit the growth of gram-positive bacteria and reduce their acetic acid production, while increasing the intestinal adhesion of gram-negative bacteria and their propionic acid production, thereby reducing CH4 production (Jayanegara et al., 2020). Saponin-rich plants can selectively regulate the rumen microbial population to enhance propionic acid production, so a decrease in the ratio of acetic acid to propionic acid is generally accompanied by reduced CH4 production (Ugbogu et al., 2019).

Effect of fenugreek on CH4 emissions.

Many other studies have reported fenugreek’s positive effects on CH4 emissions (Table 2), and no serious adverse effects of fenugreek ingestion in ruminants have been reported to date (Goel, Makkar & Becker, 2008b; Kala, 2019; Mohini, 2007). In an in vitro simulation experiment, CH4 was reduced by 15.1% when 50 mg fenugreek was added to 60 ml fermentation broth, and no significant effects were found on fermentation pH and TVFA’s concentration (Li, 2015). Kumar et al. (2016) found that a 2% fenugreek seed supplement reduced CH4 by 22.8% in an in vitro rumen simulation, improved DM and organic matter digestibility, and increased the feed degradation rate. The addition of 19, 21, 23, or 25% fenugreek to vetch-oat to in vitro rumen simulation, decreased CH4 production by 45, 35, 72, or 67%, respectively, which may be beneficial for improving nutrient utilization and animal growth (Arhabbr, Ablabr & Zitouni, 2014). The positive effects of fenugreek on CH4 emission are mainly attributed to its secondary metabolites. In addition, the content of several fatty acids, especially linoleic acid, is much higher in fenugreek than in other fodder crops, which can reduce the activity of rumen methanobacteria and change the process of biohydrogenation (Islam et al., 2017). However, Silva et al. (2021) reported that supplementation of 16 g of whole-seed fenugreek meal per day in the basal diet of dairy cows has no significant effect on rumen pH, NH3-N, TVFA’s concentration, C2:C3 ratio and enteric CH4 emissions. Rejil & Mohini (2006) also found that TVFA’s concentration, bacterial biomass, total gas as well as CH4 production were not affected after adding fenugreek extracts to the diet. The reasons for this difference may be related to the experimental environment, animal species and the dosage of fenugreek.

Table 2 Effects of fenugreek on CH4 emissions from ruminants.

Source	Animal/ organism	Doses	Effects	References	
Fenugreek seeds saponin	In vitro	11.54 mg/40 ml (buffered rumen fluid)	CH4 reduction of 1.97%; 39% reduction in protozoa; methanogens of bacteria and fungi were reduced	Goel, Makkar & Becker (2008a)	
Fenugreek	In vitro	2.5%, 5%, and 7.5% of substrate	CH4 inhibition of 19.5%, 26.1%, 35.3%, respectively	Pattanaik et al. (2018)	
Fenugreek	In vitro	(100% of fenugreek; 50:50 mixture of fenugreek and alfalfa; 100% of alfalfa) as substrate	CH4 (ml/g DM) (42.8, 52.6, 64.1); Methanogen (2, 1, 3.02, 4.23) (×104/mL); acetate: propionate (3.25,3.40,3.56); respectively	Niu et al. (2021)	
Fenugreek leaf extract	In vitro	20 mg/30 ml (buffered rumen fluid)	CH4 inhibition of 16.6% (37.3–30.1) (ml/g DM); ratio of acetate to propionate is reduced (3.39–2.85)	Dey (2015)	
Fenugreek	In vitro	19, 21, 23, or 25% of vetch-oat hay	CH4 inhibition of 45, 35, 72, or 67%, respectively	Arhabbr, Ablabr & Zitouni (2014)	
Fenugreek seeds	In vitro	66 mg/40 ml (buffered rumen fluid)	CH4 reduction of 9.7%; 56% reduction of protozoa (357–157) (×103 per ml)	Goel, Makkar & Becker (2008b)	
Fenugreek seeds	In vitro	2% of substrate	CH4 reduction of 22.8% (24.30–18.75) (ml/g DM)	Kumar et al. (2016)	
Fenugreek seeds	In vitro	1% of substrate	CH4 reduction of 34.1%	Lakhani et al. (2019)	
Fenugreek extract	In vitro	50 mg/60 ml (buffered rumen fluid)	CH4 reduction of 15.1%	Li (2015)	
Fenugreek seeds	In vitro	2% of substrate (wheat straw and concentrates mixture in the ratio of 60:40)	CH4 reduction of 19.6 (34.22–27.5) (ml/g DM)	Mohini (2007)	
Fenugreek seed extracts	In vitro	2% of substrate	CH4 production, TVFA concentration and bacterial biomass were not affected	Rejil & Mohini (2006)	
Fenugreek straw	In vivo/cull sheep	40% of diet	CH4 reduction of 26.8%	Bhatt et al. (2021)	
Whole-seed fenugreek powder	In vivo/dairy cows	16 g/cow/day	No significant difference between CH4 and acetate: propionate	Silva et al. (2021)	
Notes.

DM: dry matter.

Fenugreek in moderate doses does not have harmful effects on ruminants, but more long-term studies in vivo are needed to verify its effect and explore the specific mechanism of action.

Effects on the productivity of ruminants

Changes in the global environment (such as global warming and pandemics) and rising human demand for organic and healthful foods have necessitated greater efforts on the part of animal protein producers (livestock, poultry and aquaculture sectors) in recent years to assure adequate protein at low cost (Capper & Bauman, 2013). Plant-derived chemicals, such as fenugreeks’ bioactive compounds, have been proven in studies to increase animal performance, health, and product quality (Champa Wijekoon et al., 2020). The effects of fenugreek on ruminant growth, lactation, and antioxidant capacity will be discussed in this study (Table 3).

Table 3 Benefits of fenugreek for ruminants.

Performance	Plant Parts	Animal models	Dose/Concentration (diet)	Duration	Effects	References	
Growth	Fenugreek seed	Barbarine lamb	12% of diet	82 d	No significant on growth	Ibidhi & Salem (2022)	
	Fenugreek seed extract	Holstein dairy heifers	2 g/head/day	35 d	Improve intake, digestibility and metabolism to promote growth	Taiwo et al., (2023)	
	Fenugreek seed	Lactating Goats	50 g/d	63 d	Increase food intake by 5.9%	Cayiroglu et al. (2023)	
	Fenugreek seed	Nubian Goats	5%, 10%, 15% of diet	75 d	Improve dry matter intake, crude protein intake and nutrient digestibility	Mahala (2013)	
	Fenugreek seed	Awassi lambs	2.5 g/head/day, 5 g/head/day, 7.5 g/head/day	42 d	No significant on growth	Al-Wazeer (2017)	
	Fenugreek seed	Turkish Saanendairy goats	100 g/day	8 weeks	No significant on growth	Akbag, Savas & Yuceer (2022)	
	Fenugreek seed	Barki rams	20g /head/day or40 g/head/day	–	Dry matter intake increased with 20 g fenugreek seeds/head/ day and decrease with 40 g/head/day	Saleh (2004)	
	Whole-seed fenugreek powder	Dairy cows	16 g/cow/day	21 d	No significant on growth	Silva et al. (2021)	
	Raw fenugreek seeds	Goats	3% of digestible methionine	21 d	Improve digestibility coefficient and growth rate	Mir & Kumar (2012)	
Milk production	Fenugreek	Lactating goats	30 g/d	6 weeks	Milk production ↑37%; feed intake ↑	Hasin et al. (2019)	
	Fenugreek seed	Lactating goats	25% of diet	3 weeks	Milk production ↑67%; fat content ↑	Al-Shaikh, Al-Mufarrej & Mogawer (1999)	
	Fenugreek seed	Lactating Beetal goats	10 g/d	30 d	Milk production ↑43.9%	Sahoo et al. (2020)	
	Fenugreek seed	Goats	2 g/kg/d	–	Milk production ↑110%	Alamer & Basiouni (2005)	
	Fenugreek seed	Buffaloes	5% of diet	30 d	Milk production ↑9%	Degirmencioglu et al. (2016)	
	Fenugreek	Lactating Murrah buffaloes	50 g/d	90 d	Milk production ↑28.52%	Kirar et al. (2020)	
	Fenugreek seed	Lactating Egyptian buffaloes	150 g/d	16 weeks	Milk production ↑18%	Mahgoub & Sallam (2016)	
	Shatavari, Jivanti and Fenugreek	Lactating Kankrej cows	1:1:1 (20 g/d)	60 d	Milk production ↑24.89%	Patel et al. (2017)	
	Fenugreek seed	Surti buffaloes	1.5%	10 weeks	Milk production ↑8.87%	Choubey et al. (2018)	
	Fenugreek seed	Lactating Beetal goats	10 g/d	30 d	Milk production ↑43.85%; levels of lipid peroxidation ↓; superoxide dismutase activity ↑	Sahoo et al. (2020)	
	Fenugreek seed	Dairy goats	50 g/d, 100 g/d	3 months	Milk production ↑8.2; milk production ↑34.2%	El-Tarabany et al. (2018)	
	Fenugreek seed	Turkish Saanendairy goats	100 g/d	8 weeks	Milk fat concentration ↑; milk yield ↑	Akbag, Savas & Yuceer (2022)	
	Fenugreek seed	Lactating goats	50 g/d	63 d	Milk production ↑14.88%	Cayiroglu et al. (2023)	
	Fenugreek seed	Sudanese desert sheep	2.5 g/kg	7 weeks	Milk production ↑; weight ↑	Muddathir (2012)	
Antioxidant ability	Fenugreek seed	Buffalo calves	44 g/100 kg body weight	9 months	Improve serum glucose, globulin, antioxidants, T-4 and testosterone levels	Kumari et al. (2019)	
	Fenugreek seed	Dairy goats	50 g/d, 100 g/d	3 months	Improve physiological, hematological parameters and antioxidant capacity	El-Tarabany et al. (2018)	
	Fenugreek seed	Lactating beetal goats	10 g/d	30 d	Reduce levels of lipid peroxidation and improve the superoxide dismutase activity	Sahoo et al. (2020)	
	Fenugreek seed extract	Holstein dairy heifers	2 g/head/day	35 d	Improve antiviral immune status and reduce oxidative stress damage	Taiwo et al., (2023)	

Nutritive value of fenugreek

Fenugreek seeds contain 58% carbohydrate, 23–26% protein, 6.4% fat, 25% fiber, 6% volatile oil, 3% ash, 1.6% starch, and 0.4% soluble sugar. Fenugreek leaves contain 6% carbohydrate, 4.4% protein, and 1.1% fiber (Syed et al., 2020). Fenugreek seed protein is of good quality, with a higher content and a better amino acid composition than soy protein isolate. Additionally, compared to soy protein, it has higher denaturation temperature, foaming capacity, solubility, and stability, which makes it a useful source of protein for a range of functional meals (Wijekoon et al., 2021). Fenugreek seeds are also abundant in minerals including K, Mg, Ca, Zn, Mn, Cu and Fe; 53 elements have been isolated from fenugreek seeds, of which 11 are essential to the human diet, and their contents are in the order of Fe > Zn > Mn > Cu > Mo > Ni > V > Cr > Co > Si > Se (Srinivasan, 2006). Numerous vitamins, such as vitamins A, B1, B2, C, niacin, and nicotinic acid, are also abundant in fenugreek seeds. There are 17 main fatty acids in fenugreek, including linoleic, oleic, and linolenic acids, the majority of which are unsaturated (Table 4) (Ahmad et al., 2016).

Table 4 Nutrition composition of fresh fenugreek leaves and mature fenugreek seeds.

Nutrient composition	Fenugreek seeds	Fresh fenugreek leaves	References	
Moisture	8.84 g	86 g	Ahmad et al. (2020)	
Crude protein	23 g	4.4 g	Ahmad et al. (2020)	
Fat	6.41 g	1.1 g	Ahmad et al. (2020)	
Carbohydrate	58.35 g	6.6 g	Ahmad et al. (2020)	
Crude fiber	24.6 g	1.1 g	Ahmad et al. (2020)	
Fatty acids	1.46 g	0.9 g	Ahmad et al. (2020)	
Ca	160 mg	395 mg	Srinivasan (2006)	
Fe	14 mg	16.5 mg	Srinivasan (2006)	
K	530 mg	31 mg	Srinivasan (2006)	
Mg	160 mg	67 mg	Srinivasan (2006)	
Cu	33 mg	0.26 mg	Srinivasan (2006)	
Na	19 mg	76 mg	Srinivasan (2006)	
P	370 mg	51 mg	Srinivasan (2006)	
Vitamin A	3 µg	–	Ahmad et al. (2016)	
Thiamine	340 µg	40 µg	Ahmad et al. (2016)	
Riboflavin	290 µg	310 µg	Ahmad et al. (2016)	
Vitamin B6	0.6 µg	–	Ahmad et al. (2016)	
Vitamin C	12–24 mg	52 mg	Ahmad et al. (2016)	
Ascorbic acid	12–23 mg	52 mg	Ahmad et al. (2016)	
Folic acid	84 µg	–	Srinivasan (2006)	
Nicotinic acid	1.1 mg	0.8 mg	Srinivasan (2006)	
β-carotene	96 µg	2.3 mg	Srinivasan (2006)	
Notes.

Values expressed per 100 g.

Growth performance

Fenugreek contains a large amount of essential amino acids, such as lysine and tryptophan, which are more abundant than in conventional cottonseed- and rapeseed-meal and can fully meet the amino acid nutritional requirements for livestock growth and development (Acharya, Thomas & Basu, 2008). Mahala (2013) and Mir & Kumar (2012) reported that dietary supplementation with 3% fenugreek seed improved DMI, crude protein intake, DM digestibility and crude protein digestibility in goats, as well as growth performance. The DMI and apparent total tract digestibility of DM, crude protein, and neutral detergent fiber were greater in dairy heifers given supplementary 2 g/animal/day of fenugreek seed extract as compared to CON (Taiwo et al., 2022). According to a recent study, saponin changed the metabolome and microbial population in the rumen to promote efficient glucose metabolism (Wang et al., 2019). Fenugreek’s capacity to raise insulin sensitivity by obstructing intestinal glucose absorption may potentially be connected to the increase in food intake that it causes. The 4-hydroxy-isoleucine in fenugreek seeds can increase pancreatic insulin release in goats and regulate gluconeogenesis, thereby promoting feed intake and contributing to improved growth performance (Chen et al., 2021). The high nutritional value of fenugreek makes it a high-quality forage crop, but its anti-nutritional content cannot be ignored. Saleh (2004) observed that Barki rams (52.6 kg) had an increase in DM intake with 20 g of fenugreek seeds per head per day and a decrease with 40 g of fenugreek seeds per head per day. The threshold level of secondary metabolites determines whether they act as trophic factors, or as antinutritional factors (Champa Wijekoon et al., 2020). Adding a moderate level of saponins (2–3%) of DM improved the utilization of dietary protein, growth performance, milk yield and fertility of sheep, as well as reducing the parasite burden. However, excessive concentrations of saponins (depending on their source, type, and animal species) combine with proteins to produce complexes that inactivate digestive enzymes and decrease the digestibility of proteins (Malisch et al., 2015). Therefore, it is vital to determine the optimum level of feed supplementation for whole fenugreek, or its extracted components.

Fenugreek has also been used as a feed flavor-enhancer, it is suggested that fenugreek seeds may influence the hypothalamus gland to activate the brain’s hunger center and heighten the desire to eat (Degirmencioglu et al., 2016; Kamel, 2001). Long-term oral administration of fenugreek extract increases food intake and feeding motivation; dairy cattle appear to prefer food with added fenugreek (Degirmencioglu et al., 2016). In addition, fenugreek seeds contain diosgenin, a growth-promoting compound absent from other legumes, which can promote bovine growth through its steroidal properties, potentially replacing synthetic growth promoters (Al-Wazeer, 2017; Shim et al., 2008). Diosgenin appears to stimulate growth hormone secretion from the pituitary gland, which promotes bovine growth (Cao & Yang, 2011). Diocin, a growth hormone secretagogue, may be the cause of the group who took fenugreek supplements’ increase in body weight. Its structure is comparable to that of estrogen, which binds to pituitary cell receptors to boost GH secretion and cause a rise in body weight by releasing hormone (GH-RH).

Milk production and milk quality

Fenugreek is a widely used herb for stimulating milk production and it is effective in various mammals (Javan, Javadi & Feyzabadi, 2017). Feeding goats with 30 g of fenugreek daily increased the milk yield by 37% (Hasin et al., 2019); the daily milk yield from buffalo increased by 18% when 150 g fenugreek was added to the daily diet (Mahgoub & Sallam, 2016); goats fed 2 g/kg fenugreek daily increased their milk production by 110% and growth hormone levels increased, suggesting that more energy from feed intake was directed to growth (Alamer & Basiouni, 2005). The optimal dosage of fenugreek for stimulating ruminant milk production has not been determined.

An important factor affecting lactation is the level of prolactin (PRL). In addition, growth hormone (GH), insulin, leptin, estrogen, progesterone, oxytocin and thyroid hormone-releasing hormone also play important roles in lactation. Fenugreek may increase milk production by stimulating the secretion of endogenous hormones (Tabares, Jaramillo & Ruiz-Cortés, 2014). It was found that the PRL concentration was significantly higher in goats with 2% fenugreek added to the ration than in the control group, and it was hypothesized that the increase in milk production was mediated by stimulation of PRL (Janabi, 2012). Diosgenin in fenugreek can stimulate the secretion of PRL and the release of GH as well as promote mammary gland development and improve milk flow. PRL has estrogen-like effects and can increase milk production by reducing the secretion of dopamine (an inhibitor of prolactin secretion) in the hypothalamus (Sahoo et al., 2020). In a 7-week feeding experiment, the addition of 60 g of fenugreek seeds per day to the ration of goats increased milk production by 13% and serum growth hormone by 28.6% (Alamer & Basiouni, 2005). GH and its effector insulin-like growth factor 1 (IGF-1) have the ability to guide nutrition flow to the mammary tissue or directly stimulate the mammary gland to produce milk. It has been advised that buffaloes’ plasma growth hormone may be a potential mediator of the effects of fenugreek (Saleh, 2004). The lactogenic action of fenugreek may also be attributed to the phytoestrogen flavonoid compounds in its seeds, which are structurally similar to the mammalian hormone, 17 β-estradiol and can exert a weak estrogenic effect. Phytoestrogens affect the gonadal and growth axes of the endocrine system, regulating the growth, development and reproduction of mammals, as well as stimulating the proliferation of estrogen receptor-positive human breast cells, changing the structure of the mammary gland and regulating the endocrine system (Cayiroglu et al., 2023; Choubey et al., 2018; Sreeja, Anju & Sreeja, 2010). Fenugreek contains a precursor of niacin which participates in the formation of niacinamide adenine dinucleotides (NAD+) (Sevrin et al., 2019), which is a cofactor for both PARP (a DNA repair enzyme) and sirtuins (longevity proteins). NAD+ has an important function in cell survival and transcriptional regulation (Braidy et al., 2014). The fenugreek alkaloid trigonelline increases the NAD+ content of the mammary gland, extending the lifespan and enhancing the function of mammary gland cells, thereby promoting lactation. Hasin et al. (2019) reported that the non-protein amino acid, 4-hydroxy-idoxuridine in fenugreek seeds increases glucose-induced insulin secretion from goat islet cells, so fenugreek intake extends the mid-lactation peak by stimulating insulin secretion and regulating the insulin/GH/IGF-1 axis. However, fenugreek may be less effective when animals are undernourished, have mammary gland dysplasia, or have hormone dysregulation (El-Tarabany et al., 2018).

Consuming fenugreek also influences the components of milk, such as cholesterol, fatty acids, and lactose, by promoting the expression of genes related to the uptake of glucose (Glut1), the production of galactose (Pgm1, Ugp2), and lactose (B4galt1, Lalba). Additionally, it enhances fluid flow from the mammary epithelial cells into the breast secretory vesicles and increases the expression of key milk proteins (Csn2, Lalba) and an amino acid transporter (Slc7a5), thereby increasing milk production and lactose content (Kirar et al., 2020; Sevrin et al., 2020). In addition, dietary fenugreek reduces the cholesterol content of cows’ milk, increases the concentration of functional fatty acids (linoleic acid, linolenic acid and conjugated linoleic acid) (Shah & Mir, 2004), and improves the flavor and palatability of the milk. Similarly, Akbag, Savas & Yuceer (2022) reported that the addition of fenugreek seeds to cow diets increased the concentration of unsaturated fatty acids in milk, but the concentration of linoleic and linolenic acids in milk did not increase with the addition of fenugreek. This could be because the rumen hydrogenates the majority of these fatty acids.

Management of oxidative stress

Oxidative stress has a great impact on the health and productivity of livestock and poultry, decreasing disease resistance, survival rate, feed reward, growth performance and increasing stunting of young animals. Fenugreek contains abundant flavonoid compounds, especially vitexin, isovitexin and other polyphenols, which function as antioxidants and stimulate the oxidative stress response, both of which scavenge free radicals and ameliorate the oxidative damage they cause to cells (Ruwali et al., 2022). Sahoo et al. (2020) found that goats fed 10 g of fenugreek seeds daily had decreased lipid peroxidation, increased superoxide dismutase activity and improved antioxidant status. This is because of polyphenols acting on the cellular antioxidant signaling pathway, activating related transcription factors and regulating the expression of downstream genes. Under heat stress, total antioxidant capacity and antioxidant enzyme activity were significantly increased and fat peroxidation levels were significantly reduced in goats after ingesting 100 g of fenugreek seeds per day, indicating that fenugreek can effectively alleviate oxidative stress brought about by high-temperature (El-Tarabany et al., 2018). Taiwo et al. (2022) also reported that gene set transcript levels associated with erythrocyte uptake/release of CO2, release/uptake of oxygen, and O2/CO2 exchange in erythrocytes of cows fed a corn silage-based diet plus 2 g/animal/d of fenugreek seed extract were reduced (FDR ≤ 0.05), indicating a decrease in oxidative stress. It’s mainly due to the ability of saponins to scavenge superoxide by forming H2 peroxide intermediates, thus preventing biomolecular damage caused by free radicals (Naidu et al., 2011). In addition, fenugreek contains abundant polyunsaturated fatty acids, which increase the activity of antioxidant enzymes, relieve oxidative stress in mammals and may also improve reproductive performance. Treatment of postpartum buffalo with a Chinese herbal mixture, including fenugreek, accelerated uterine recovery, promoted early uterine cleaning, decreased the duration of the service period and the delay to first insemination, and increased reproductive efficiency (Japheth et al., 2021).

Conclusions and recommendations

CH4 released by ruminants is a major contributor to greenhouse gas emissions and energy loss, and in the upcoming years, ruminant production systems will confront even greater difficulties. As a result, methods to reduce CH4 production while preserving animal health and productivity are being developed and researched. The profound influence of rumen fermentation on ruminant nutrition, food production and the environment has stimulated extensive research into managing the rumen microbiome to maximize productivity and reduce the environmental impact of ruminant production; dietary intervention to achieve these goals is a promising approach. When it comes to controlling the rumen flora and reducing the generation of CH4, fenugreek and its secondary metabolites—such as saponins, tannins, and flavonoid compounds—have shown inconsistent outcomes during rumen fermentation. However, most studies have shown that the addition of fenugreek or its bioactive compounds to the basal feed reduces CH4 production and improves rumen fermentation. In addition, fenugreek, as a dietary additive, can promote animal growth and increase the utilization potential of feed. Its rich nutrient content and the relative abundance of the growth promoter, diosgenin can improve the daily weight gain of beef cattle and sheep, and shorten the achievement of a commercially-acceptable weight. Dietary fenugreek improves milk yield and quality, mainly mediated by flavonoids and trigonelline, which affect the endocrine system and hormone secretion and ameliorate oxidative stress. Therefore, the livestock feed industry has the opportunity to develop natural, environmentally-friendly feed additives based on fenugreek.

This review has identified some gaps and weaknesses in past fenugreek research, which should be addressed by future research, i.e.: (1) Studies on the use of fenugreek to reduce CH4 emissions from ruminants were mostly performed by short-term in vitro simulations; long-term in vivo trials will be needed to confirm the efficacy of fenugreek under real conditions; (2) the mechanisms of action by which fenugreek secondary metabolites influence the composition and metabolic activity of rumen bacteria are poorly understood and further research into effective strategies for CH4 emission reduction is needed; (3) the efficacy of fenugreek in promoting lactation and improving growth performance, and its toxicological safety is influenced by factors such as dose, animal species and strain, and environmental conditions, so future research should aim to improve understanding of these factors. Future research should also aim to improve understanding of the potential benefits of fenugreek, explore the mechanisms of action of its various components and functions, and clarify the optimal level and formulation of fenugreek supplementation, to provide a theoretical basis for further development of its application in ruminants.

We thank International Science Editing for editing this manuscript.

Additional Information and Declarations

Competing Interests

Author Contributions

Data Availability

Shijun Liu is employed by Shanghai Mugao Biotechnology Co., Ltd. The authors declare there are no competing interests.

Xiangbiao Zeng conceived and designed the experiments, performed the experiments, authored or reviewed drafts of the article, and approved the final draft.

Yiwen Chen analyzed the data, prepared figures and/or tables, and approved the final draft.

Wenjuan Li conceived and designed the experiments, authored or reviewed drafts of the article, and approved the final draft.

Shijun Liu conceived and designed the experiments, authored or reviewed drafts of the article, and approved the final draft.

The following information was supplied regarding data availability:

This is a literature review.

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
