# Peer review of "Application of fenugreek in ruminant feed: implications for methane emissions and productivity"

_PeerJ, doi:10.7717/peerj.16842_

## Round 0.1 · original submission · Major Revisions

Please provide a point-by-point rebuttal to all of the comments and suggestions from the reviewers.

Reviewer 1 ·

Basic reporting

The manuscript include information about the benefits of applying fenugreek in ruminant feed to reduce methane emissions and productivity. However, the title does not show which part of the crop should be included in ruminant feed (seeds or vegetative parts or both?)
Line 77: Authors did not mention about which group of bioactive compounds are important to reduce green house gas (GHG) emission in ruminants.
Line 93-101: There are numerous research related to use of fenugreek as a forage crop. Authors need to include more of these recent literature information in the introduction.
Line 151, The secondary metabolites of fenugreek were extensively studied and more information is available related to GHG emission and different forage types. Therefore, I recommend to expand this section with more literature to explain why fenugreek is important. For example, if saponins are mentioned, the specific chemical groups or compounds in fenugreek need to be mentioned as well.
Reference section needs to be re- formatted and some references (eg: Line 514) were listed incorrectly.

Experimental design

I have following suggestions to improve the manuscript
It is advisable to create a separate section on use of fenugreek as a forage crop. Some varieties and cultivars of fenugreek are grown in North America as forage.
I also recommend addition of another figure as a summary on the benefits on fenugreek for ruminants based on published literature.
The tables listed are not informative enough. I suggest to add a table with fenugreek bioactive molecules (different part of plants) and their roles/potentials identified based on previous evidence.

Validity of the findings

The literature cited in identifying phytochemicals/bioactive molecules in fenugreek should be improved and the information presented is not enough to validate and support the written content..

Additional comments

Overall the manuscript is informative but it lacks some valuable information on fenugreek to use as a feed additive to reduce GHG emission..

Reviewer 2 ·

Basic reporting

The review explores the use of fenugreek to reduce ruminal methane production and improve animal performance. The review is broad and cross-disciplinary interest and fails within the scope of the journal. Many reviews on phytogenic feed additives showed their ability to reduce methane emissions from ruminants. However, a review focusing on fenugreek is not common.
The Introduction section adequately introduces the subject and makes it clear to readers.

Experimental design

The article content is within the Aims and Scope of the journal, and a rigorous investigation was performed to a high technical & ethical standard. Because this is a review article, methods of collecting data and information are shortly described. Sources are adequately cited and paraphrased as appropriate. The review is well organized logically into coherent subsections.

Validity of the findings

Conclusions are well stated, linked to the original research question, and limited to supporting results.

Additional comments

I have some additional minor comments:
1. Please avoid using long sentences. I noted a lot of them throughout the manuscript.
2. Figure 1 is not clear. Please draw it with a high resolution.

Reviewer 3 ·

Basic reporting

The manuscript is well-written using clear language. The introduction and background effectively reflect the objective of the paper. The authors have met the standards set by Peerj for a review paper and have used the latest references to explain the objective in detail. Overall, the paper is well-drafted, despite a few language errors. However, there are some errors in the references section, such as incorrect page numbers for some articles (line 461), inconsistent writing of "in vivo" and "in vitro," and "Bubalus bubalis" (lines 491, 531, and 519). Therefore, it is recommended that the authors follow the rules and regulations of Peerj when citing references.

Experimental design

The paper's contents fall within the scope of the journal. Adequate literature from various online databases has been collected and concisely summarized. References are properly cited throughout the paper for clarity.

Validity of the findings

There is a significant variation in the amount of fenugreek, either in the form of fodder or seed that is fed to ruminant animals for a specific purpose. The author did not include the cost-effectiveness of feeding fenugreek to reduce methane production and improve animal productivity. The conclusions are clearly stated in terms of feeding levels and briefly justify the objective of the review paper.

Additional comments

Line 26, 111: Check the spelling of datebases
Line 124: Use omasum and abomasums instead of flap and crumpled stomach
Line 128-29: Rewrite the sentence for unambiguous understanding
Line 136-38: Rewrite the sentence And during fermentation…..for clear-cut understanding
Line 185: How is bacteriolytic activity? Justify for proper understanding
Line 200, 202: Verify spelling of bacteria Butyrivibio fibrisolvents
Line 220, 228: Do not repeat the name of the scientist; write only the year of publication in brackets throughout the review paper.
Line 245, 254: What is the significance of numbers 67 and 11 in the respective line?
Line 284: Use in vivo in italic as in vivo
Line 334: Include as (Al-Wazeer, 2017; Shim et al., 2008).
Line 350-52: Rewrite sentence And growth hormone (GH), insulin, leptin, estrogen, progesterone…
Line 536-538: Remove it as the same reference is repeated in line 619-621.
Table 1: Write in vitro or in vivo in italic

---

## Round 0.2 · accepted · Accept

The authors have addressed the suggested reviewers' comments in a satisfactory way.